# Enhancing the Photo and Thermal Stability of Nicotine through Crystal Engineering with Gentisic Acid

**DOI:** 10.3390/molecules27206853

**Published:** 2022-10-13

**Authors:** Devin J. Angevine, Kristine Joy Camacho, Javid Rzayev, Jason B. Benedict

**Affiliations:** 1Department of Chemistry, University at Buffalo 730 Natural Sciences Complex, Buffalo, NY 14260-3000, USA; 2Department of Chemistry, University at Buffalo 838 Natural Sciences Complex, Buffalo, NY 14260-3000, USA; 3Department of Chemistry, University at Buffalo 826 Natural Sciences Complex, Buffalo, NY 14260-3000, USA; 4Department of Chemistry, University at Buffalo 771 Natural Sciences Complex, Buffalo, NY 14260-3000, USA

**Keywords:** crystal engineering, solid-state, nicotine salt, nicotine, gentisic acid

## Abstract

The use of crystal engineering to convert liquids into crystalline solids remains a powerful method for inhibiting undesired degradation pathways. When nicotine, a liquid sensitive to both light and air, is combined with the GRAS-listed compound, gentisic acid, the resulting crystalline solid, exhibits enhanced photo and thermal stability. Despite a modest ΔT_m_ of 42.7 °C, the melting point of 155.9 °C for the nicotinium gentisate salt is the highest reported for nicotine-containing crystalline solids. An analysis of the crystal packing and thermodynamic properties provides context for the observed properties.

## 1. Introduction

There is an evolving need to stabilize volatile and sensitive chemicals that exist as liquids at ambient conditions [1,2]. In comparison to solids, liquid-state chemicals may present particular hazards with respect to their safe handling and storage [3]. Of particular interest is the stabilization of substances intended for human consumption such as active pharmaceutical ingredients (APIs). A multitude of drug candidates exist as liquids at ambient conditions that exhibit some form of instability such as sensitivity to moisture or UV light sources [4].

To stabilize these drug candidates, pharmaceutical formulations are developed which convert the liquid APIs into crystalline solids. One approach, co-crystallization, combines an API with a selected molecular partner (i.e., a coformer) through robust non-covalent interactions. Co-crystallization has been demonstrated as a method to stabilize volatile compounds, modulate select properties [5,6], and inhibit API degradation [7,8].

One API that is starting to garner industrial and academic interest is nicotine [9,10,11,12,13,14]. Nicotine is one of the most widely consumed drug compounds across the globe, yet pure nicotine, an oily liquid under ambient conditions, exhibits an array of degradation pathways upon exposure to air, light or excessive temperatures [15,16]. Upon exposure to ultraviolet (UV) radiation sources, nicotine degrades into relatively benign products such as nicotinic acid as well as potentially harmful degradation products such as diradical methylene which may cause undesirable biological effects [17].

The ability to isolate nicotine in the solid-state through co-crystallization was demonstrated in 2017 by Capucci et al. [18] wherein they “tamed” nicotine by isolating it through co-crystallization using several halogenated coformers. However, the selection of these particular coformers rendered the obtained materials unsuitable for human consumption. Utilizing coformers or salt formers which are suitable for human consumption presents the prospect of isolating nicotine as a solid, crystalline material with improved degradation properties over pure nicotine and improved safety over nicotine materials containing halogenated compounds.

To create the targeted safer materials, US Food and Drug Administration (FDA) generally recognized as safe (GRAS) listed compounds and other food safe molecules such as flavoring agents can be utilized as coformers or salt formers [19,20]. GRAS listed compounds are substances that the US FDA has rigorously tested and approved as being safe for human consumption [21,22]. Herein, 2,5-dihydroxybenzoic acid (gentisic acid) was selected as it is a common metabolite of drug compounds such as aspirin and it is also found naturally in many fruits such as kiwis and apples [23]. In addition, gentisic acid was selected as it possesses a pKa1 value of 2.53. As expected, protonation occurs at the pyrrolidine of the nicotine (pK_a_ = 8.02, ΔpK_a_ = 5.49) as opposed to the pyridyl group (pK_a_ = 3.12, ΔpK_a_ = 0.59) [24]. In this work the synthesis, single crystal X-ray diffraction (SC-XRD) structure, photostability, and thermal characterization of a salt comprised of the starting materials nicotine and gentisic acid is discussed (Figure 1). These results are compared to previously published nicotinium salts that were created utilizing the GRAS listed salt former malic acid and orotic acid, as well as the halogenated nicotine co-crystal counterparts.

## 2. Results & Discussion

### 2.1. SC-XRD Analysis

As seen in Figure 1, when combined in a 1:1 ratio and evaporated from methanol, (S)-nicotine and gentisic acid crystallize to form a 1:1 salt (S)-nicotinium gentisate (CCDC deposition number 2168648). That a salt is formed within the lattice is supported by the presence of a q-peak (0.55 e^−^/Å^3^) near N1 corresponding to the transferred proton. Additionally, the C1-O2 and C1-O1 bond lengths of 1.2554(15) Å and 1.2754(14) Å, respectively, are consistent with the presence of a carboxylate group. Within the asymmetric unit there are two hydrogen bonding interactions of note (Appendix A). An intramolecular S(6) type motif occurs within the gentisate salt former resulting in a 2.5483(14) Å interaction. The second is observed between the gentisate carboxylate and the pyrrolidyl nitrogen of the (S)-nicotinium wherein a discrete (D-type graph set) motif is observed resulting in a 2.6775(13) Å hydrogen bonding interaction. This type of hydrogen bonding between the carboxylate of the salt former and the pyrrolidyl nitrogen of the nicotinium is similar to those found in the malate salts and the orotate salt.

To form the bulk structure, infinite 1-D chains of the type *C*(7) are formed between the hydroxyl group in the 5-position of a gentisate molecule and the carboxylate group of an adjacent gentisate. This creates 1-D wires consisting of asymmetric unit blocks running along [100]. These wires stack along [010] forming sheets which consequently stack along c* with each subsequent sheet being rotated 180° about c* (Figure 2).

### 2.2. Hirshfeld Surface Analysis

Hirshfeld surfaces are a critical tool that can assist in the understanding of a variety of properties, such as the electrostatic potential, shape index or the d_norm_ of a crystal system [25,26,27]. Through the evaluation of such properties, information may be ascertained with regard to the type and quantity of interactions present for an API. Such data can then be utilized to engineer additional systems involving the targeted API [28].

The interaction environment of the API (S)-nicotinium was analyzed for the (S)-nicotinium gentisate salt described herein. This allowed for a direct comparison of the API interaction environment across several previously reported nicotinium salts, including the family of (S)-nicotinium malate salts [CCDC REF codes: QAXQOZ (Malate I), QAXQOZ01 (L-Malate II), QAXQOZ02 (DL-Malate), and QAXRAM(D-Malate)], as well as the (S)-nicotinium orotate hemihydrate salt (CCDC REF code: PEKXAI) [14,29,30]. As seen in Figure 3, in the fingerprint plots of the Hirshfeld surface a small central spike corresponding to H-H interactions was observed as peaking around 1.1 Å d_i_ (internal distance) × 1.1 Å d_e_ (external distance). Peaking around 0.7 Å d_i_ × 1.0 Å d_e_, a large spike corresponding to the H-O/O-H interactions was observed. Two smaller peaks consistent with the N-H/H-N type interactions were observed as peaking around 1.0 Å d_i_ × 1.4 Å d_e_ and 1.4 Å d_i_ × 1.0 Å d_e_. Two larger wings were observed which corresponded the more distant C-H/H-C interactions. The remaining C-C, C-O/O-C, and N-O/O-N interactions were all relatively small and distant interactions as indicated by the fingerprint plots. The gentisate salt API environment possessed 16.7% H-O/O-H type interactions, 49.9% H-H type interactions, 20.5% H-C/C-H type interactions, 10.2% H-N/N-H type interactions, 0.8% O-C/C-O type interactions, 1.3% C-C type interactions, and 0.6% N-O/O-N type interactions.

Like all reported nicotinium salts, H-H interactions were the most abundant type surrounding the API (Figure 4). Significantly fewer H-O/O-H type interactions were observed in the gentisate compared to the other salts, likely due to the gentisate hydroxyl group on the five-position interacting very strongly with adjacent gentisate molecules thus distancing itself too far from the nicotinium to significantly contribute. The H-N/N-H type interactions were similar in abundance to each of the malate salts. The abundance of H-C/C-H interactions were almost double that of the L-malate I and DL-malate salts and around 8% more than the L-malate II and D-malate salts. The remaining three types of interactions found in the gentisate API environment (O-C/C-O, C-C, and N-O/O-N) were in relatively low abundance and distant across all of the reported salts. Details pertaining to the quantity of each interaction around the API for the gentisate and all other previously reported salts can be found in Appendix A.

### 2.3. Thermal Properties

As observed in the nicotinium malate salts and nicotinium orotate hemihydrate salt, it has been shown that nicotine thermal properties in the solid-state can be modulated through the selection of a salt former [29,30]. Through the understanding and application of the thermal properties of an API, further materials may be rationally designed with targeted properties [31]. To assess the thermal properties of the synthesized salt, four replicates of the material were analyzed on a Stuart SMP10 melting point apparatus and consequently analyzed via differential scanning calorimetry (DSC). Approximately ten-gram sample was scanned twice from 0 °C to 200 °C and back to 0 °C each time (Figure 5). An endothermic transition associated with the anticipated melting event was observed as peaking at 155.9 °C. Integration of the endotherm yielded an enthalpy of fusion ($\Delta H_{fusion}^{{}^{\circ}}$) of 29.97 kJ mol^−1^. This led to an entropy of fusion ($\Delta S_{fusion}^{{}^{\circ}}$) value of 6.986 × 10^−2^ kJ mol^−1^ K^−1^ when computed using the Gibbs free energy equation (Appendix A). This salt exhibits a glass transition during the second DSC scan as the material is amorphous after the endothermic transition in the first scan and does not recrystallize.

These thermodynamic properties were then compared to the halogenated nicotine co-crystals and the nicotinium malate salts to assess the overall performance of this newly synthesized salt. The nicotine co-crystals reported by Capucci et al. possess a range of melting points between 54 °C to 92 °C, much lower than the 155.9 °C melting point observed for the system described herein [18]. Additionally, the nicotinium malate salts possessed melting points ranging from 93.4 °C to 122.2 °C, with enthalpies of fusion ranging between 11.22 kJ mol^−1^ and 18.62 kJ mol^−1^ [29]. Thus, nicotinium gentisate represents one of the highest melting nicotine solids reported in peer-reviewed literature.

The ΔT_m_ property has previously been established as a performance statistic for the halogenated nicotine co-crystals by Capucci et al. and the nicotinium malate salts. This value is derived by taking the salt melting point and comparing it to that of the salt former melting point (T_m_), with the difference being reported as ΔT_m_ [18]. Like the malate salts, the gentisate salt reported herein exhibits an increased ΔT_m_ between the non-API salt former and the synthesized crystalline salt. As the salt former gentisic acid has a melting point of 203.1 °C, the (S)-nicotinium gentisate salt possesses a ΔT_m_ value of 47.2 °C [32]. This is lower than the nicotine containing co-crystals reported by Capucci et al. which had a reported range of 55–78 °C [18]. The gentisate did, however, possess a ΔT_m_ value that was greater than any of the nicotinium malate salts as they possessed a ΔT_m_ range of 5.0–33.8 °C [29]. However, the orotate salt had a much larger ΔT_m_ value of 215.4 °C [30].

### 2.4. Photostability Analysis

It has been well established that pure (S)-nicotine is not stable upon exposure to ultraviolet (UV) irradiation. Some degradation products such as nicotinic acid may not be as harmful as other degradation products such as methylamine [33]. It has been demonstrated previously that the transformation of nicotine into nicotinium salts effectively eliminates the formation of undesired products upon prolonged irradiation [14,29,30].

The photostability of (S)-nicotinium gentisate was analyzed by exposing a sample to a broad-spectrum UV irradiation light source for 24 h. After 24 h had passed, a representative portion of the sample was taken and analyzed via ^1^H-NMR. The spectrum acquired post irradiation was then compared to a ^1^H-NMR spectrum acquired from a portion of the sample taken prior to UV irradiation. The spectra were then compared to check for any observable photodegradation. This was also completed for the API nicotine and the salt former to inspect their photostability.

As a control, a sample of pure (S)-nicotine was stored in the dark with no UV exposure for 24 h. ^1^H-NMR spectra were taken from the sample of (S)-nicotine before after 24 h of being in the dark. This control had little to no observable degradation in the spectra. A separate sample of pure (S)-nicotine was also exposed to 24 h of UV irradiation, which led to a multitude of new peaks being observed in the post irradiation spectrum when compared to the spectrum acquired prior to irradiation (Appendix A). There was no detectable degradation in the spectrum of the acid salt former (Appendix A) acquired after UV irradiation when compared to the sample taken prior to the UV irradiation, while pure liquid nicotine did demonstrate the expected degradation as many new peaks were observed (Appendix A). Meanwhile, the gentisate salt had no observable degradation in the ^1^H-NMR spectrum after the UV irradiation (Figure 6). Phase purity of the salt was confirmed via PXRD analysis both before and after the irradiation period (Appendix A). This demonstrates that the solid-state can offer protection against UV induced degradation of (S)-nicotine. This falls in line with the photodegradation testing results of the nicotinium malate salts and the orotate salt, wherein no detectable products of photodegradation were observed.

## 3. Materials and Methods

### 3.1. Materials

(S)-nicotine (98%) and gentisic acid (98%) were purchased from Alfa Aesar (Ward Hill, MA, USA) and Combi-Blocks (San Diego, CA, USA), respectively. Methanol (HPLC grade 99.9%) and n-heptane (99%) were each purchased from Fisher Scientific (Fair Lane, NJ, USA). Dimethyl sulfoxide-D6 (D, 99.9%) and Methanol-D4 (D, 99.8%) were purchased from Cambridge Isotope Laboratories Inc. (Andover, MA, USA)

### 3.2. Salt Synthesis

Gentisic acid (770.6 mg, 5.0 mmol) was added into a 20 mL scintillation vial. Methanol (6.0 mL) was added with vigorous agitation. (S)-Nicotine (0.8 mL, 5.0 mmol) was added via micropipette in the dark to avoid degradation. The resulting solution was capped and vortexed for 30 s at 3000 rpm on a VWR Mini Vortexer MV I. The solution was then stored in the dark uncapped to allow for crystal formation while the solvent slowly evaporated. Once the solvent evaporated, the crystalline product was collected via vacuum filtration, washing with n-heptane (3 × 5 mL) (1.472 g, 93.05%). The yield was computed based upon the formula weight of (S)-nicotinium gentisate (F.W. 316.4 g/mol).

### 3.3. X-ray Diffraction (XRD)

X-ray diffraction data were collected using a Bruker SMART APEX-II CCD diffractometer (Bruker AXS, Billerica, MA, USA) installed at a rotating anode source (MoKα radiation, λ = 0.71073 Å) and equipped with an Oxford Cryosystems (Cryostream700; Oxford Cryosystems Ltd, Long Hanborough, England) nitrogen gas-flow apparatus. Five sets of data (360 frames each) were collected by the rotation method with 0.5° frame-width (ω scans) with a 2.0 s exposure time for the single crystalline sample. The sample was run at 90 K. Using Olex2, the structure was solved with intrinsic phasing via the ShelXT structure solution program and refined with the ShelXL software suite (George M. Sheldrick, Göttingen, Lower Saxony, Germany) using least squares minimization [34,35,36]. A q-peak (0.55 e^−^*/*Å^3^) was located 0.838 Å from N1 and was assigned as a proton consistent with charge transfer from the carboxylic acid of the salt former. The atomic coordinates of H atoms attached to heteroatoms were freely refined with thermal parameters constrained to be U_iso_(H) = 1.5U_eq_(N) or 1.5U_eq_(O). H atoms connected to carbon atoms were placed geometrically (C–H = 0.95 Å) and refined with thermal parameters constrained to be U_iso_(H) = 1.2U_eq_(C). Images of the structures were created using Olex2 and the CSD: Mercury Visualization and Analysis of Crystal Structures software suite (Clare F. Macrae, Cambridge, UK) [37]. Absolute configuration was assigned based upon the stereochemistry of the API (S)-nicotine. The Flack parameter was removed in accordance with IUCr standards for reporting a structure in which the absolute configuration is assigned based upon a known reference molecule, which in this structure is (S)-nicotinium.

### 3.4. Crystal Melting Points

A Stuart SMP10 melting point apparatus was utilized to measure the melting point of the synthesized compounds. 4 replicates were run for the salt.

### 3.5. Differential Scanning Calorimetry

A differential scanning calorimeter, model DSC Q200 (TA Instrument, New Castle, DE, USA) was used to measure the thermal transitions of the sample. About 10 mg of the synthesized compound was placed into an aluminium pan and sealed. The salt was scanned from 0 °C to above the melting point observed on the Stuart SMP10 at 20 °C/min under argon and nitrogen flow (10 mL/min. each) for 2 full cycles. Exothermic transitions are displayed as positive heat flows. The enthalpy of fusion was computed from the integrated area under the curve and the entropy of fusion computed by applying the Gibbs’ free energy equation in accordance with equilibrium between the solid and liquid state being achieved at the fusion point.

### 3.6. UV Photodegradation

NMR analysis was performed on a sample of the salt former orotic acid, (S)-nicotine, and the synthesized (S)-nicotinium orotate hemihydrate salt. Each sample was then irradiated with ultraviolet (UV) light in a home-built vented box with air flow for 24 h using four Southern New England Ultraviolet Company RPR—3000A (Branford, CT, USA) UV bulbs (λ = 300 nm). NMR analysis was then carried out on each sample to screen for any UV photodegradation of products.

### 3.7. Nuclear Magnetic Resonance (NMR)

NMR analysis was carried out using a Bruker NEO 400 MHz NMR (Bruker AXS, Billerica, MA, USA) spectrometer equipped with an iProbe, AutoTune assembly with variable temperature control, as well as a SampleCase autosampling unit. Eighty transients were run for each sample. Appropriate deuterated solvents are labeled in each spectrum.

### 3.8. Hirshfeld Surface Analysis

The Hirshfeld surface of (S)-nicotinium orotate hemihydrate and the previously synthesized nicotinium malate salts was generated using Crystal Explorer 17.5 [38]. For all salts the d_norm_ surface was mapped using the color scale with the range −0.050 a.u. (red) to 0.600 a.u. (blue). In addition, 2-D fingerprint plots were generated as the outer nuclei (d_e_) versus the inner nuclei (d_i_) using an expanded interaction distance ranging from 0.6 Å to 2.8 Å.

### 3.9. Powder X-ray Diffraction (PXRD)

Powder X-ray diffraction data were collected using a Rigaku Ultima IV X-ray Diffraction (XRD) System equipped with standard attachment (CuKα radiation, λ = 1.54 Å). Data collection was performed over the 2θ range from 2° to 45° utilizing a 0.02° incremental step. A scanning speed of 5° per minute was utilized. Slit heights were set as follows: divergence slit: 2/3°; divergence height limiting slit: 10mm; scattering slit: 2/3°; receiving slit: 0.3mm. PXRD patterns were simulated using the Mercury 4.0 2021.2.0 visualization and analysis of crystal structures software suite (Clare F. Macrae, Cambridge, UK) [37]. A 0.02° step was utilized for the simulated PXRD pattern along with a full width half max of 0.1. All PXRD patterns were normalized to a maximum intensity of 10,000 counts.

### 3.10. Infrared (IR) Spectroscopy

Infrared spectral analysis was carried out on a Perkin Elmer (Waltham, MA, USA) Spectrum Two FTIR spectrometer equipped with an attenuated total reflectance (ATR) module. Eight scans were averaged for each spectrum.

## 4. Conclusions

The novel nicotine salt (S)-nicotinium gentisate was successfully synthesized using slow evaporation from methanol. This salt exhibited distinctive discrete hydrogen bonding interactions within the asymmetric unit. Infinite 1-D *C*(7) chains are formed between the hydroxyl group in the 5-position of a gentisate molecule and the carboxylate group of an adjacent gentisate. Thus, 1-D wires are formed consisting of asymmetric unit blocks running along [100]. These wires stack along [010], forming sheets which consequently stack along c* with each subsequent sheet being rotated 180° about c*.

DSC measurements determined that the salt melts at 155.9 °C and possesses an enthalpy of fusion (${\Delta H}_{\mathrm{fusion}}^{{}^{\circ}}$) of 29.97 kJ mol^−1^. This melting point temperature was higher than any of the reported nicotine containing co-crystals or nicotinium malate salts. Similar to all previously reported solid forms of nicotine, (S)-nicotinium gentisate had a significantly increased melting point in comparison to that of pure (S)-nicotine (−79 °C). The gentisate salt described herein also possessed a ΔT_m_ value that was larger than any of the nicotinium malate salts, but not as large as the nicotine co-crystal ΔT_m_ value that have been reported.

No detectable photodegradation products were observed after prolonged irradiation of the nicotinium gentisate salt. This further supports the supposition that the crystallization of liquid nicotine is a viable means of minimizing and even eliminating potentially harmful product formation arising from undesirable nicotine degradation.

## Data Availability

Deposition number 2168648 contains the crystallographic data for this paper. These data are provided free of charge by the joint Cambridge Crystallographic Data Centre and Fachinformationszentrum Karlsruhe Access Structures service www.ccdc.cam.ac.uk/structures (accessed on 9 September 2022).

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
