# Peer review of "Enhancing the Photo and Thermal Stability of Nicotine through Crystal Engineering with Gentisic Acid"

_molecules, 2022, doi:10.3390/molecules27206853_

Round 1

Reviewer 1 Report

The paper reports the design and characterization of a nicotinium salt, aiming at stabilizing a nicotin crystalline form. The rationale is clear and the characterization is careful. I suggest to accept after minor revisions:

Page 2, line 67: please characterize H-bonds by distances and angles; truncate the appropriate number of significant digits

Figure 1 caption: there is no need to use the symmetry descriptors; I suggest to move the entire nicotinium cation so that the H-bond is found in the asymmetric unit.

Page 8, line 228: Flack, not flack

Page 9, sections 3.3 and 3.8: it is not reported if H atoms were normalized; if not, the calculations should be redone.

Reviewer 2 Report

Angevine, et al. reported a salt form of nicotine with gentisic acid, which turns the liquid nicotine to a crystalline solid with a melting point as high as 155.9 °C. Meanwhile, the light sensitivity of the nicotine was very well overcome after such salt formation. In addition to the structural analysis of both crystal packing and Hirshfeld surface, thermodynamic properties of the salt was investigated via DSC experiments. This work provides a practically significant example to improve the stability of the widely used drug compound nicotine. However more experiments are required for the new pharmaceutically active complex. Therefore, this reviewer suggests the publication of this work in Molecules after major revision by considering the following comments.

1.       The title is “Enhancing the Photo and Thermal Stability of Nicotine Through Crystal Engineering with Gentisic Acid”. How the thermal stability of nicotine was enhanced? Does the pure nicotine decompose easily by heating? It needs the thermal stability characterization for the pure nicotine as well to say so.

2.       The phase purity is also very important to the pharmaceutically active compounds. Then a powder X-ray diffraction experiment should be performed for the nicotinium gentisate salt before and after the UV irradiation.

3.       The crystal structure was perfectly solved. A general description of the treatment for the hydrogen atoms is necessarily given in the experimental part. Particularly, the reviewer suggests the authors to clearly demonstrate how the hydrogen atom between the gentisate carboxylate O2 and the nicotinium pyrrolidyl N1 is located. This is crucial to name the complex an adduct or a salt. Although no doubt about the structure from the good quality data, IR is suggested to be done for both the “salt” and its pure components, which is also helpful to confirm the complex as a salt.

4.       For the DSC experiments for the salt, the thermal behavior in the second run is different from the first. What’s does it mean? What is the state of the salt after cooling down from melting? A hot stage microscopy measurement is suggested.

5.       Does the temperature or the UV irradiation affect the position of the hydrogen between O2 and N1? IR measurements for the sample after melting and UV irradiation are also necessary.

6.       What about the solubility of the salt in water and buffer solutions of different pH?

7.       Table 1 was mentioned in the text in page 5 line 143, but the table cannot be found in the manuscript.

Reviewer 3 Report

The manuscript entitled" Enhancing the Photo and Thermal Stability of Nicotine Through Crystal Engineering with Gentisic Acid" is an interesting study in the area of crystal engineering. Authors should address few comments before the manuscript is accepted for publication.

i) It is not very clear from the crystal structure that Nicotine and Genstistic acid hydrogen bonded complex is salt/cocrystal. The authors should give the C=O bond distances of carboxylate functional group in the manuscript. 

ii) IR spectra of acid and salt will be useful to predict the formation of salt/cocrystal. 

iii) the delta pKa values of acid and amines should be given in the manuscript. Why pyrrolidyl nitrogen is protonated as compared with pyridyl nitrogen? Explain based on pKa values. 

iv) The dissolution study of newly synthesized nicotine salt will be useful for its application for its real life application. 

Round 2

Reviewer 2 Report

All the issues have been addressed by the authors. This reviewer now is happy with it and recommend the publication in Molecules.